# Protein-Bound Uremic Toxins in Senescence and Kidney Fibrosis

**DOI:** 10.3390/biomedicines11092408

**Published:** 2023-08-28

**Authors:** Yi Yang, Milos Mihajlovic, Rosalinde Masereeuw

**Affiliations:** 1Division of Pharmacology, Utrecht Institute for Pharmaceutical Sciences, Utrecht University, 3584 CG Utrecht, The Netherlands; y.yang@uu.nl; 2Entity of In Vitro Toxicology and Dermato-Cosmetology, Department of Pharmaceutical and Pharmacological Sciences, Vrije Universiteit Brussel, 1090 Brussels, Belgium; milos.mihajlovic@vub.be

**Keywords:** chronic kidney disease, uremic toxins, renal tubular transport, extracellular matrix remodeling, apoptosis resistance, inflammatory response, senescence-associated secretory phenotype factors

## Abstract

Chronic kidney disease (CKD) is a progressive condition of kidney dysfunction due to diverse causes of injury. In healthy kidneys, protein-bound uremic toxins (PBUTs) are cleared from the systemic circulation by proximal tubule cells through the concerted action of plasma membrane transporters that facilitate their urinary excretion, but the endogenous metabolites are hardly removed with kidney dysfunction and may contribute to CKD progression. Accumulating evidence suggests that senescence of kidney tubule cells influences kidney fibrosis, the common endpoint for CKD with an excessive accumulation of extracellular matrix (ECM). Senescence is a special state of cells characterized by permanent cell cycle arrest and limitation of proliferation, which promotes fibrosis by releasing senescence-associated secretory phenotype (SASP) factors. The accumulation of PBUTs in CKD causes oxidative stress and increases the production of inflammatory (SASP) factors that could trigger fibrosis. Recent studies gave some clues that PBUTs may also promote senescence in kidney tubular cells. This review provides an overview on how senescence contributes to CKD, the involvement of PBUTs in this process, and how kidney senescence can be studied. Finally, some suggestions for future therapeutic options for CKD while targeting senescence are given.

## 1. Introduction

Kidney fibrosis leads to organ failure by an excessive accumulation of extracellular matrix (ECM), which is the common endpoint for a variety of progressive chronic kidney diseases (CKD) [1]. Senescence is a special form of permanent cell cycle arrest, which limits proliferation and is highly related to inflammation and fibrosis. Senescent cells exacerbate these processes by releasing senescence-associated secretory phenotype (SASP) factors, which are of pro-inflammatory and profibrotic nature [2]. Uremic toxins are metabolites that accumulate during kidney disease. Protein-bound uremic toxins (PBUTs) are mostly less than 500 Da but are poorly removed with kidney dysfunction, as they are tightly bound to plasma proteins and can also hardly cross dialyzer membranes [3,4]. PBUTs, such as indoxyl sulfate (IS) and p-cresol sulfate (PCS), accumulate in CKD, maintaining and reinforcing CKD and kidney fibrosis [5,6]. Recent studies reported that IS and PCS activate the renal RAAS/TGF-β pathway and induce epithelial mesenchymal transition (EMT) [6]. EMT is a common process during fibrosis and concerns the loss of a differentiated epithelial-like state of cells (e.g., cell-to-cell junctions) to acquire a more mesenchymal-like phenotype (e.g., enhanced ECM expression) [7]. Senescence and EMT are both characterized by cell dedifferentiation, loss of epithelial phenotype, cell cycle arrest, and negative effects on surrounding cells [8]. IS triggers senescence [9] and induces EMT with ECM (i.e., α-SMA) deposition in vitro [10], which suggests that PBUTs may induce kidney fibrosis by propagating senescence. However, the crosstalk between PBUT-related fibrosis and senescence-related fibrosis remains unclear. It is worth noting that EMT can be induced by toxins, such as IS, in vitro in epithelial cells, but such an EMT mechanism in vivo in renal fibrosis is still questionable, as pericytes, endothelial cells, and bone marrow-derived stem cells may be sources of myofibroblasts [11,12,13]. Here, we provide some mechanistic insight into how PBUTs could promote kidney fibrosis by accelerating senescence both in vitro and in vivo. 

## 2. The Mechanisms of Kidney Fibrosis 

Kidney fibrosis is induced by the abnormal accumulation of ECM, which often initiates as the result of a wound healing response. The response is orchestrated by complex activities of different cells, including macrophages and T cells, epithelial cells, myofibroblasts, and endothelial cells. Four major phases are involved in this process: (1) primary injury that initiates a fibrotic response; (2) the activation of effector cells, triggering the fibrosis signaling (e.g., TGF-β signaling); (3) production of ECM; and (4) deposition of ECM that promotes tissue fibrosis and eventually leads to kidney failure [1]. 

### 2.1. Main Signaling of Fibrosis

Three main signaling pathways are involved in fibrosis: transforming growth factor (TGF)-β, wingless/Int (WNT), and yes-associated protein (YAP)/transcriptional coactivator with PDZ-binding motif (TAZ) signaling pathways [14]. TGF-β signals through both canonical (Smad-based) and non-canonical (non-Smad-based) pathways; Smad-based TGF-β signaling plays a central role in the development of renal fibrosis; non-Smad-based profibrotic actions of TGF-β signaling are regulated by interactions with other signaling pathways (e.g., MAPK/ERK and PI3K/AKT pathways signaling) [15]. The WNT signaling pathway is activated by secreted lipid-modified proteins of the WNT family. Activation of WNT signaling stabilizes β-catenin; the nuclear translocation of β-catenin initiates the transcription of fibrotic genes, such as collagen and fibronectin [16,17]. YAP and TAZ are major players of the Hippo pathway, which is involved in organ development, epithelial homeostasis, tissue regeneration, wound healing, and immune modulation; ECM stiffening promotes the nuclear activity of YAP/TAZ, which in turn promotes the development of a fibrotic cellular phenotype, including increasing the expressions of the connective tissue growth factor (CTGF) and plasminogen activator inhibitor 1 (PAI-1) [18,19,20]. These three signaling pathways show a cross-talk during fibrosis. Their mechanisms range from modulating the availability of growth factors and the availability of membrane-bound receptors to nuclear entry and activation of transcription factors [14]. Recent studies revealed that TGF-β and WNT signaling are also related to senescence [21,22].

### 2.2. ECM in Kidney Fibrosis

The ECM is a non-cellular component of tissue that provides essential structural support for cellular constituents and acts as an active component in cell signaling. It is composed of water, proteins, and polysaccharides and is responsible for cell–cell communication, cell adhesion, and cell proliferation [23,24]. There are two main types of ECMs: interstitial connective tissue matrix (e.g., collagen I and fibronectin) and the basement membrane (e.g., collagen IV and laminins) [25]. The interstitial connective tissue matrix is responsible for tissue structure, while the basement membrane underlies or surrounds most tissues, including epithelial and endothelial tissues, and interacts with cells (Figure 1) [25,26]. Three histologically distinct compartments with a variety of ECMs are affected in kidney fibrosis: the glomeruli, tubulointerstitium, and vasculature (Table 1) [27]. As a result of ECM remodeling, the deposition of matrix proteins is observed in kidney fibrosis (Table 1). 

### 2.3. ECM Remodeling 

ECM remodeling is referred to as a balance between degradation and production of ECM. When the balance is disrupted [27], a positive feedback loop resulting in increased ECM production drives the development of fibrosis [29]. The cleavage of ECM by different proteases is the main process during the remodeling and includes matrix metalloproteinases (MMPs), adamalysins, meprins, and metalloproteinase inhibitors (reviewed in [25]). MMPs are the main enzymes involved in ECM degradation and remodeling. MMPs can cleave ECM components and activate other MMPs and proteins. Various cytokines (interleukin [IL] and tumor necrosis factor [TNF]) and growth factors (epidermal growth factor [EGF] and transforming growth factor [TGF]) may be involved in the gene expression of MMPs at the transcription level [30]. Adamalysins include disintegrin, metalloproteinases (ADAMs), and ADAMs with a thrombospondin motif (ADAMTS); adamalysins contain twenty-one ADAMs and nineteen ADAMTS proteins; shedding of various substrates, including adhesion ligands, growth factors, and their receptors; and cytokines [31]. Meprins are the only astacin proteinases that can be bound to membranes or secreted as soluble factors; meprin subunits cleave a variety of biologically active peptides, many cytokines, and chemokines, leading to an alteration in the biological functions/activities of those factors/proteins [32]. The tissue inhibitors of metalloproteinases (TIMP) are endogenous inhibitors of MMPs and adamalysins. Each TIMP specifically binds to their target MMPs or adamalysins, regulating the production/deposition of various ECM components, such as collagens, fibronectins, and laminins [33].

## 3. Senescence 

Senescence is a special form of permanent cell cycle arrest, which limits cellular proliferation. It was first reported as a loss of replicative capacity in cultured human fibroblasts in 1961 [34]. Senescent cells are currently regarded as a potentially important contributor to different types of diseases [35], including aging-related diseases [36], kidney disease [37], and pulmonary disease [38]. Some senescent cells can be cleared by immune cells through the chemo-attracting of immune cells, followed by tissue regeneration, which is called acute (short-term) senescence, while chronic (long-term) senescent cells accumulate and create a lesion, aggravating the pathology [39,40]. Major types of senescence are highlighted as replicative senescence (RS), oncogene-induced senescence (OIS), and stress-induced (premature) senescence (SIS) (Figure 2). RS is linked to telomere shortening that is associated with cell division. This type of senescence is a consequence of activating a DNA damage response (DDR), which is induced by short telomeres through the induction of the cell cycle inhibitor p21, arresting proliferation [41,42,43,44]. 

Oncogene-induced senescence refers to cell cycle arrest by the aberrant activation of oncogenic signaling, which promotes the initiation and development of cancer [45]. This can be caused by numerous oncogenes, including constitutively active variants in the RAS/MAPK pathway (RAS-induced senescence), as well as in the PI3K/AKT pathway (AKT-induced senescence). The former undergoes a DDR, while the latter is independent of DDR [46]. Stress-induced (premature) senescence appears after exposing cells to chemical or physical stresses, including radiation waves, hydrogen peroxide, and chemotherapeutic agents [47], leading to cellular stress, increased reactive oxygen species (ROS) generation, and subsequent DNA damage, eventually contributing to senescence [40,47].

### 3.1. Mechanisms of Senescence 

As discussed, senescence is triggered by various stressors, including DNA damage, mitochondrial dysfunction, metabolism, and cell stress [2,48,49]. Most of them accompany the DDR outcomes, followed by activation of the cell cycle arrest and the release of SASP factors [50,51]. 

#### 3.1.1. Cell Cycle Arrest

Cell cycle arrest in senescence is largely mediated via the p53/p21^CIP1/WAF1^ (p21) and p16^Ink4a^ (p16)/pRb checkpoint pathways controlled by DDR [52,53], which are independent processes in senescence induction. p53/p21 is activated when DDR occurs, promoting a p21-dependent G0/G1 cell cycle arrest [54,55]. p16 suppresses retinoblastoma 1 (pRb) and prevents the actions of the cyclin-dependent kinases, which induces a G1 cell cycle arrest [56]. Acute DNA damage drives cell cycle arrest via the p53/p21 pathway, while chronic DNA damage followed by the induction of the p16/pRB pathway maintains cell cycle arrest and senescence [57]. As a key mediator of cell cycle arrest, some studies also demonstrated that p21 can be upregulated via a p53-independent mechanism [58,59]. Checkpoint signaling pathways are associated with p53-mediated apoptosis [60]. During DDR, the abnormal expression of p53 may further lead to apoptosis resistance.

#### 3.1.2. Apoptosis Resistance 

Senescent cells are resistant to apoptosis [61] via intrinsic and extrinsic pathways. The intrinsic pathway refers to the mitochondrial pathway of apoptosis, related to mitochondrial outer membrane permeabilization (MOMP) [62]. In this pathway, MOMP and the release of cytochrome c are required to trigger apoptosis, and it involves Bcl-2 and caspase family proteins [63,64]. The Bcl-2 family is divided into three main groups: anti-apoptotic (Bcl-2, Bcl-xl, and Mcl-1), pro-apoptotic (Bax and Bak), and pro-apoptotic BH3-only (Bim, Bid, Bad, and Puma) proteins [65]. The balance between pro-apoptotic and anti-apoptotic Bcl-2 family members determines the threshold in MOMP for apoptosis. Caspase proteins are downstream players of MOMP in the intrinsic apoptosis pathway [66]. After the activation of Bax–Bak-dependent MOMP, cytochrome c is released from the mitochondria, stimulating the activation of caspase-9 and its downstream executioners, caspases-3 and -7, to initiate apoptosis [64]. The extrinsic pathway is initiated via death receptors that bind death ligands secreted by other cells (e.g., macrophages and natural killer cells), activating caspase-8 and its downstream executioner, caspases-3, to initiate apoptosis [62]. Natural ligands, including TNF, Fas-L, and TRAIL, are known to bind to their receptors, TNFR1, TNFR2, Fas, and TRAIL-R, to activate caspase-8 [67]. Caspase-8 activation can lead to the cleavage of Bid to tBid and initiates the mitochondria-mediated intrinsic apoptosis pathway [62]. Accumulation of dysfunctional mitochondria in senescent cells has been reported [68]. Senescent cells are in a primed apoptotic state, triggered by the abnormal regulation of anti-apoptotic and pro-apoptotic Bcl-2 family proteins, keeping cells alive without undergoing proliferation or apoptosis [69]. SASP factors, such as TNF-α [70,71], released from senescent cells also play a role in the extrinsic apoptosis pathway. This kind of regulation finally inhibits the activation of executioner caspase-3, leading to apoptosis resistance and chronic senescence.

#### 3.1.3. SASP Factors

SASP factors are related to a DDR and are generally proinflammatory and/or profibrotic compounds, including numerous cytokines, chemokines, growth factors, and matrix-metalloproteinases (MMPs) [2,72]. Several reports described that SASP factors are not only responsible for the maintenance and reinforcement of senescence but also key players during its transmission [73]. Cytokines, such as IL-6 and IL-8, are well-proven to play such critical roles in stress-induced senescence [74,75,76]. IL-6 maintains senescence through the p53/p21 pathway [77,78]. This role of IL-6 in senescence is shared by IL-8, which is expressed as a function of IL-6 [75]. Both cytokines are regulated by IL-1α [79]. The nucleotide-binding oligomerization domain (NOD)-like receptor 3 (NLRP3) inflammasome is upregulated in senescence, which leads to expressions of IL-1α and IL-1β, resulting in the upregulation of SASP factors and the reinforcement of senescence in a paracrine manner [80]. Chemokine signaling is also reported as being responsible for reinforcing growth arrest by the CXCR2 receptor and CXCR2-binding chemokines [74]. Chemokines, including CCLs and CXCLs, are involved in stress (radiation)-induced senescence, thus leading to fibrosis [81]. Chemokine signaling also plays a role in OIS; senescent cells increase the survival of cancer cells via CXCL12/CXCR4 signaling, leading the collective invasion in thyroid cancer [82]. Growth factors such as CTGF and TGF-β induce senescence and are accompanied by the upregulations of IL-6 and IL-8, thus reinforcing paracrine senescence [83,84]. TGF-β induces CTGF expression through the activation of Smad3 and p53 [85,86], inducing cell cycle arrest and contributing to senescence [87]. Accumulation of MMPs is also observed in senescence [88]. MMPs shed ectodomains of cell surface receptors and activate other SASP factors, hence promoting senescence via paracrine signaling [89].

### 3.2. Senescence and Fibrosis

Senescence contributes to fibrosis in multiple organs [90,91,92] and is considered to be a result of the release of SASP factors and the pathways triggered by them (Figure 3). TGF-β signaling controls cell proliferation and survival, regulating apoptosis and senescence [87], and initiates fibrosis through the canonical Smad signaling and Smad-independent signaling pathways, with subsequent ECM deposition [93]. CTGF is the effector molecule of TGF-β in the kidney [94,95] and has been shown to contribute to TGF-β signaling through the extracellular signal-regulated kinase (ERK), ADAM17, ribosomal S6 kinase 1 (RSK1), and the CCAAT/enhancer-binding protein β (C/EBPβ) signaling pathway in human epithelial cells [85,96]. CTGF is necessary for the TGF-β-induced phosphorylation of Smad1 and Erk1/2, but it is not needed for the activation of the Smad3 pathway [97].

Proinflammatory mediators such as IL-1β and IL-6 are also involved in fibrosis. IL-1β augments TGF-β1-induced EMT through MAPK signaling pathways [98], which may be dependent on IL-17A [99]. IL-6 shifts acute inflammation into a chronic fibrosis state by regulating MMPs and the TGF-β pathway [100,101]. MMPs release ectodomains of cell surface receptors and activate other SASP factors [89], thus regulating ECM production and promoting EMT and kidney fibrosis [25]. For example, in fibroblasts, IL-6 promotes the expression of collagen I and stimulates the activation of TGF-β in signal transducers and activators in a transcription 3 (STAT3)-dependent manner, thus regulating MMP1, TIMP-1, and the production of collagen I; on the other hand, TGF-β promotes IL-6 production through phosphoinositide 3-kinase (PI3K) and MAPK signaling pathways [102]. Other SASP factors such as CCL2 and PAI-1 are also important players in fibrosis, exerting their effects through chemokine and TGF-β signaling, respectively [103,104].

## 4. Protein-Bound Uremic Toxins Promote Fibrosis by Accelerating Senescence

Uremic toxins are endogenous metabolites that are excreted into the urine through glomerular filtration and active transport by the proximal epithelial cells [105]. In kidney disease, uremic toxins management is compromised, which leads to the systemic accumulation of the toxins and activation of inflammation and oxidative stress. Furthermore, uremic toxins can induce profibrotic effects, promoting the progression of kidney damage [106]. Uremic toxins are divided into three distinct groups: (1) small water-soluble compounds (molecular weight <500 Da, e.g., creatinine, urea, and uric acid); (2) middle molecules (peptides with molecular weight >500 Da, e.g., IL-6, IL-8 and TNF-α); and (3) protein-bound uremic toxins (PBUTs; molecular weight mostly <500 Da, e.g., indoxyl sulfate, p-cresyl sulfate, and p-cresol) [105,107]. Small water-soluble compounds are hydrophilic, which pass through the glomerular barrier and can be removed easily by dialysis [108,109]. Most of middle molecules are peptides and difficult to remove in the process of dialysis unless the dialyzer pore size is large enough [110]. PBUTs are removed by proximal tubule cells in healthy kidneys through active secretion involving transporter proteins but poorly removed with kidney dysfunction [111]. Current dialysis therapy is limited because of the high binding to plasma proteins, with albumin being the primary carrier protein, and only a small free fraction is available for transfer across dialyzer membranes [3,4]. 

PBUTs accumulate systemically but also in kidney tissue, where they can induce oxidative stress and stimulate the production of inflammatory factors, which might be a trigger for fibrosis [112]. PBUTs induce ROS production and enhance oxidative stress and IL-1β (SASP) expression in kidney proximal tubule cells [113]. Furthermore, it has been reported that PBUTs induce TGF-β and WNT signaling, which promote ECM remodeling [114,115]. IS and PCS induce EMT by activating the renal TGF-β signaling [6] and contribute to ECM remodeling by upregulating MMP2 and MMP9 in an EGF receptor-dependent manner [116]. As discussed, TGF-β and WNT signaling are also related to senescence [21,22], which might suggest that PBUTs can be drivers of senescence and kidney fibrosis.

The accumulation of PBUTs occurs in a time- and stage-dependent manner during CKD. As the loss of kidney function in CKD is progressive and irreversible, advanced CKD has more severe uremic toxin plasma levels [5] that potentially can induce senescence [9,117]. Chronic senescence promoted by external factors (e.g., ionizing radiation, exposure to toxins, or heat stress) may develop in a time-dependent manner [118,119,120]. This suggests that PBUTs may also promote senescence time dependently during CKD, progressing the disease and reinforcing fibrosis (Figure 4). In CKD animal models, it was shown that the accumulation of PBUTs correlated with fibrosis outcome and/or senescence phenotype (Table 2). We, therefore, hypothesize that PBUTs may promote kidney fibrosis by accelerating senescence, possibly via mitochondrial dysfunction, cell cycle arrest, and the production of SASP factors.

### 4.1. PBUTs Accelerate Senescence via Mitochondrial Dysfunction 

Different types of senescence have been reported to increase ROS and mitochondrial dysfunction [68], which influences the intrinsic apoptosis pathway by the abnormal expression of Bcl-2 family and caspase family markers, which in turn maintain and reinforce senescence [133]. Overproduction of ROS during cell stress leads to mitochondrial dysfunction after kidney injury [134], which is promoted by PBUT accumulation [135]. A cocktail of PBUTs, consisting of IS, PCS, indoxyl-β-glucuronide, p-cresyl glucuronide, indol-3-acetic acid, hippuric acid, kynurenic acid, and l-kynurenine, have been shown to promote ROS production and to upregulate IL-6 in proximal tubule epithelial cells [113,136]. In addition, ROS-induced senescence was shown to require the mammalian target of rapamycin (mTOR) activation [137], and accumulated IS promoted renal fibrosis via mTOR under CKD conditions [126]. Furthermore, the class I PI3K signaling regulates and activates mTOR [138]. PCS activates NADPH oxidase through a mechanism that involves PI3K signaling, inducing ROS production and TGF-β1 secretion [139]. Interestingly, the activation of mTOR is related to renal autophagy, which is a special process for eliminating abnormal cells [140]. Dysregulated autophagy is known to be a major factor in the pathogenesis of renal fibrosis and related kidney diseases [140], involving both the tubulointerstitial compartment and glomeruli, and may also contribute to the accumulation of chronic senescent cells. PBUTs, such as IS, PCS, and hippuric acid, influence apoptosis by causing imbalances in caspase-3, caspase-9, Bcl-2, and Bax in hepatocytes, with marked ROS generation and mitochondrial damage [141]. Although there is a lack of evidence showing that PBUTs inhibit apoptosis, IS and PCS increase the expression of the anti-apoptotic genes Bcl-2, Bcl-xl and Bax in proximal tubule cells [142], which is also observed in senescent cells [61,120]. 

### 4.2. PBUTs Accelerate Senescence via Cell Cycle Arrest 

Cell cycle arrest is necessary for the repair of DNA damage after injury [143], which generally occurs in senescence and is a critical factor for fibrosis development [144]. DDR is a cause of cell cycle arrest mediated by the p53/p21 and p16/pRb pathways [51]. ROS triggers DDR, and DDR promotes ROS production by activating its downstream effectors, including p53 and p21 [145]. Recent research has suggested that PBUTs may accelerate senescence via cell cycle arrest and inhibition of cell proliferation [146,147]. Others have suggested that PCS and IS upregulate p21 and increase the number of cells positive for senescence-associated beta-galactosidase [9]. IS also promotes p53 expression, stimulating the expression of TGF-β1 and ECM deposition [148]. 

### 4.3. PBUTs Accelerate Senescence via SASP Factors 

During CKD progression, the released inflammatory (SASP) factors activate different pathways and initiate various processes, including senescence and EMT, in tubular epithelial cells [146,149]. As discussed, PBUT accumulation-induced inflammation might be one reason for senescence development. SASP factors such as IL-6, TGF-β1, and CXCL10 were reported to be increased in proximal tubule cells after the treatment with the PBUTs IS and PCS [142]. Furthermore, ROS overproduction can activate the NLRP3 inflammasome, which cleaves pro-caspase-1 and pro-interleukin-1β (IL-1β) into the proinflammatory factors caspase-1 and IL-1β, thus promoting fibrosis [150]. A cocktail of PBUTs (IS, PCS, indoxyl-β-glucuronide, p-cresyl glucuronide, indol-3-acetic acid, hippuric acid, kynurenic acid, and l-kynurenine) has been shown to promote the NLRP3 inflammasome-mediated IL-1β production via oxidative stress and NF-κB signaling [113]. Interestingly, the NLRP3 inflammasome/IL-1β also promotes cellular senescence [151]. SASP factors reinforce senescence and induce senescence transmission or paracrine senescence, which is regulated by the inflammasome [80]. Therefore, PBUTs may play an important role during CKD to promote paracrine senescence and senescence transmission. 

**Figure 4 biomedicines-11-02408-f004:**
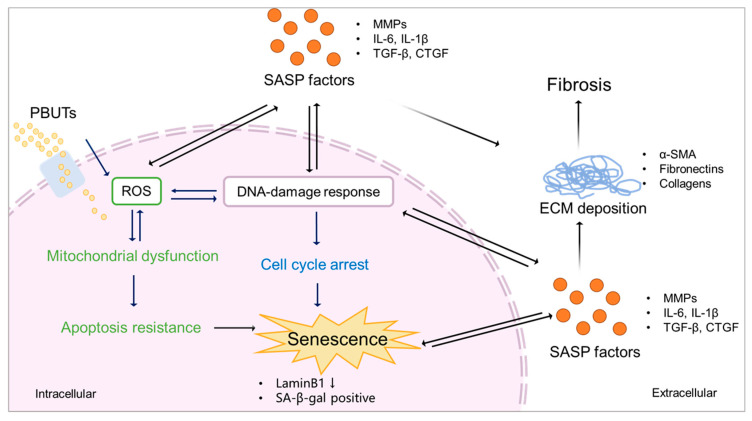
Proposed scheme of PBUTs promoting kidney fibrosis by accelerating senescence. After reaching the cells, PBUTs promote ROS production, triggering DDR and mitochondrial dysfunction, inducing apoptosis inhibition, cell cycle arrest, and the production of SASP factors, thus promoting senescence. Senescent cells show a downregulation of LaminB1 and SA-β-gal activity. SASP factors expressed by senescent cells promote ECM deposition, leading to kidney fibrosis.

## 5. Conclusions and Future Therapeutic Perspectives

PBUTs may promote senescence in CKD through the release of SASP factors (e.g., IL-6 and IL-1β) and common senescence markers (e.g., p21 and Laminb1) and trigger oxidative stress, possibly causing mitochondrial dysfunction, promoting an inflammatory response and increased resistance to cell death. As SASP factors are typically profibrotic and proinflammatory mediators, a novel treatment strategy of CKD could be inhibiting the related signaling, thus suppressing SASP expression. Potential novel agents already exist for this, including anti-fibrotic agents (e.g., TGF-β inhibitor and pirfenidone) and anti-inflammatory agents (e.g., the anti-TNF-α monoclonal antibody and infliximab) [152]. Moreover, the mTOR inhibitor rapamycin is also recognized as an SASP inhibitor (called senomorphic), reducing the development of cellular senescence [153], which could represent possibilities for CKD treatment as well [154]. In addition, strategies to inhibit senescence phenotypes by promoting cell cycle process and cell death signaling, such as an inhibitor of p21 and/or promoter of caspase proteins, could be treatment options. Considering that chronic senescent cells cannot be cleared by the immune cells, strengthening the immune system by increasing the binding affinity of the involved membrane receptors is another approach to more efficiently clear senescent cells [8]. Advanced cell therapy may be employed to specifically target senescent cells by recognizing appropriate antigens [39]. Finally, identifying and targeting most relevant and specific senescence-associated markers by means of gene therapy could be a valid approach to be investigated in the future for ameliorating kidney senescence [155]. 

## Figures and Tables

**Figure 1 biomedicines-11-02408-f001:**
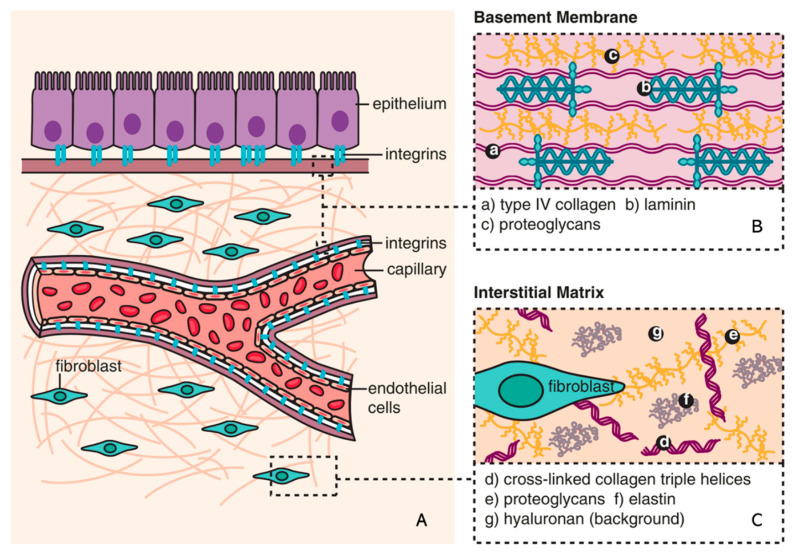
Composition of ECM (reproduced with permission from [28]). (**A**) The basic subdivision of the ECM into the (**B**) basement membrane and (**C**) interstitial matrix is shown along with major structural components (collagen and elastin), as well as the background matrix made up of proteoglycans and hyaluronan [28].

**Figure 2 biomedicines-11-02408-f002:**
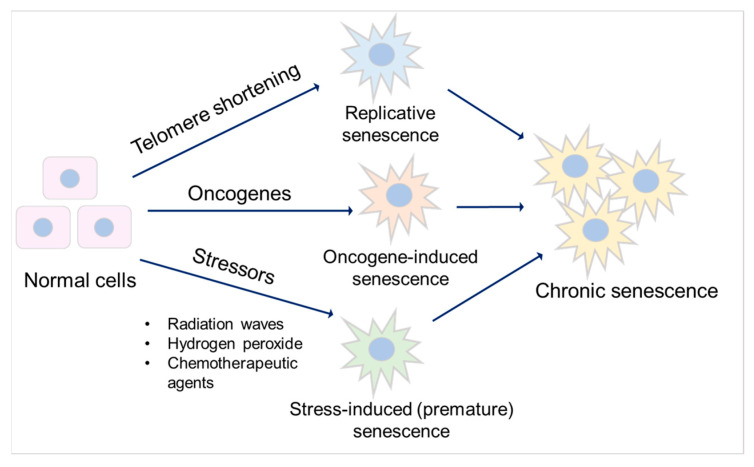
Major types of senescence. Three main types of senescence are identified. Replicative senescence links to telomere shortening that is associated with cell division. Oncogene-induced senescence refers to cell cycle arrest by the aberrant activation of oncogenic signaling, which promotes the initiation and development of cancer. Stress-induced (premature) senescence appears after exposing cells to chemical or physical stresses. Accumulation of long-term senescent cells leads to chronic senescence.

**Figure 3 biomedicines-11-02408-f003:**
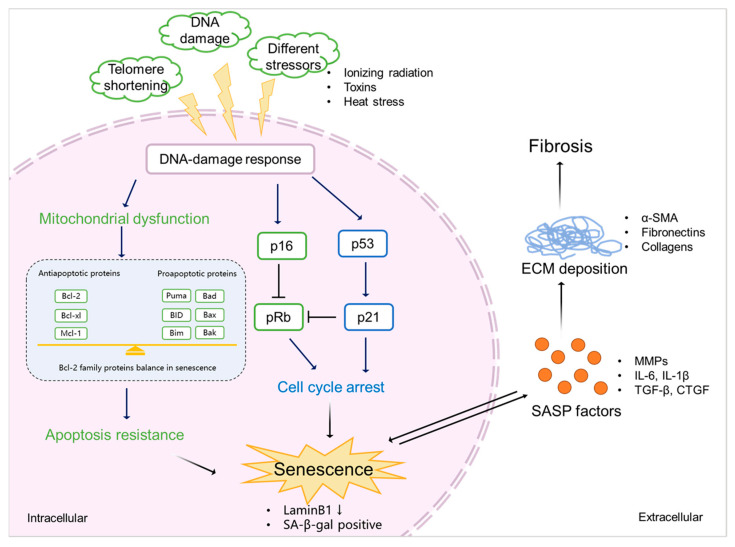
The mechanism of senescence in kidney fibrosis. Senescence is initiated by various stimulations (e.g., ionizing radiation, exposure to toxins, and heat stress), which triggers DDR. This, on the one hand, induces mitochondrial dysfunction, resulting in the abnormal expression of Bcl-2 family proteins, eventually leading to apoptosis resistance and the promotion of senescence. On the other hand, DDR mediates cell cycle arrest via p53/p21 and p16/pRb checkpoint pathways, which also results in senescence. Senescent cells show a downregulation of LaminB1 and SA-β-gal. SASP factors, including profibrotic cytokines (TGF-β and CTGF), proinflammatory cytokines (IL-6 and IL-1β), and ECM-remodeling proteases (MMPs) expressed by senescent cells promote ECM deposition (α-SMA, fibronectins, and collagens), finally leading to kidney fibrosis.

**Table 1 biomedicines-11-02408-t001:** ECM in kidney fibrosis (adapted from [27]).

Compartment	ECM in Healthy Kidney	Increased ECM in Kidney Fibrosis
Glomeruli	Mesangial Matrix: collagen IV, V, fibronectin, nidogen, laminin.	Nodular mesangial sclerosis: collagen I, III, IV, V, fibronectin, nidogen, laminin, decorin, biglycan.
	Glomerular basement membrane: collagen I, III, VI, IV, VII, XV, XVII, agrin, perlecan, nidogen, laminin.	Focal segmental glomerulosclerosis: collagen III, IV, heparan sulfate proteoglycans.
	Bowman’s capsule: collagen IV, laminins, nidogen, heparan sulfate proteoglycans.	Thickening of glomerular basement membrane: collagen I, III, VI, IV, VII, XV, XVII, perlecan, nidogen, laminin.
		Bowman’s capsule: collagen IV and heparan sulfate proteoglycans.
Tubulointerstitium	Tubular basement membrane: collagen IV, agrin, perlecan, laminin.	Thickening of tubular basement membrane: collagen IV, perlecan;
	Interstitium: collagen I, II, III, V, VI, VII, XV, fibronectin, biglycan, decorin, versican.	Interstitial fibrosis: collagen I, II, III, V, VI, VII, XV, fibronectin, biglycan, decorin, versican.
	Capillary basement membrane: N/A.	Thickening and multilayering of capillary basement membrane: N/A.
Vasculature	Intima with internal elastic lamina: elastin, perlecan, agrin, collagen XVIII, versican, biglycan, decorin.	**Neointima: versican, collagen XVIII, agrin, perlecan.**
	Media with external elastic lamina: collagen I, III, XVII, elastin, agrin, perlecan, decorin, versican.	Intima with internal elastic lamina: elastin, perlecan, agrin, collagen XVIII, versican.
	Adventitia: collagen I, III, fibronectin, elastin.	Media with external elastic lamina: elastin, collagen XVII, agrin, perlecan, versican.
		Perivascular fibrosis (thickening of adventitia): N/A.

**Table 2 biomedicines-11-02408-t002:** Overview of CKD animal models that reported PBUT accumulation, fibrosis outcome, or senescence phenotype.

CKD Model	Species	PBUTs	Fibrosis/EMT Markers	Senescence Markers/SASP Factors	Involved Pathways/Mechanism	Reference
Aristolochic acids-induced	Mouse	PCS, IS	α-SMA, collagen I, α-1 and IV	NR	TGF-β signaling	[121]
Adenine-induced	Mouse	NR	Collagen (Masson staining)	p21, Il-6, and Il-1β	chronic inflammation	[122]
5/6 nephrectomy	Rat	NR	Collagen (Masson staining)	TNF-α, IL1β, and IL-6	p38 MAPK/NF-κB signaling pathway	[123]
Ischemia-reperfusion injury	Mouse	NR	Collagen (Masson staining), fibronectin, and α-SMA	SA–β-gal, p16, p19, p53, p21, MMP-7, PAI-1, and TGF-β1	WNT and TGF-β signaling	[21]
Adenine-induced	Mouse	PCS, IS, and hippuric acid	Collagen (Masson staining) and α-SMA	NR	gut microbiota	[124]
IS-injected mouse and unilateral nephrectomy	Mouse	IS	ZO-1, occludin, claudin-1, and claudin-2	TNF-α, IL-1β, and IL-6	mitochondrial dysfunction and mitophagy impairment	[125]
Adenine-induced	Mouse	IS	α-SMA, E-cadherin, and collagen I	TNF-α and IL-6	mTOR activation	[126]
Adenine-induced	Rat	IS	fibronectin, collagen I, α-SMA, vimentin, and E-cadherin	NR	EMT	[127]
Unilateral ureteral obstruction (UUO)	Mouse	IS	Collagen (Masson staining), α-SMA, and collagen I, fibronectin, vimentin, and E-cadherin	TGF-β1	EMT	[128]
Adenine-induced	Rat	PCS, IS, hippuric acid, p-cresyl glucuronide, and indol-3-acetic acid	NR	TGF-β1	TGF-β signaling	[129]
Adenine-induced	Mouse	PCS	NR	TNF-α and IL-6	NLRP3 inflammasome pathway	[130]
Adenine-induced	Mouse	PCS, IS and p-cresyl glucuronide	collagen α-1 type 1	TGF-β1, TNF-α, MCP-1 and IL-6	production of uremic toxins and inflammation	[131]
Unilateral nephrectomy	Mouse	PCS	NR	p38 and IL-1β	Oxidative stress and inflammation	[132]
5/6 nephrectomy	Rat	Hippuric acid	α-SMA, vimentin, and collagen I	MMP9 and TIMP1	Oxidative stress and TGF-β signaling	[114]

NR—Not Reported.

## Data Availability

Data sharing not applicable.

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
