# Peer review of "Protein-Bound Uremic Toxins in Senescence and Kidney Fibrosis"

_biomedicines, 2023, doi:10.3390/biomedicines11092408_

Round 1

Reviewer 1 Report

Dear editors:  

 It is a great honor and pleasure for me to be invited as the reviewer for this important work entitled “Protein-bound uremic toxins in senescence and kidney fibrosis”. Yi Yang et al. comprehensively reviewed the patho-mechanism of protein-bound uremic toxins (PBUTs) induced cell senescence and the kidney fibrosis (glomeruli, tubulointerstitium, and vasculature). This study topic is interesting and important, attributing to Prof. Rosalinde Masereeuw’s long-term efforts and contributions in this scientific field. I have two minor comments concerning this study:

1.     Line 283, 303, 313: The “question mark” in the subheading should be deleted.

2.     Line 342 and 350: “NF-κB inhibitor resveratrol…” and “chimeric antigen receptor (CAR) T cell therapy” were not reviewed in the article that should be rephrased in the conclusion and future therapeutic application.

        Thank you for giving me the opportunity to review this interesting article. After minor revision, this important review article should be published as soon as possible.

Author Response

We thank the reviewer for appreciating our work and for providing suggestions for improvement. We deleted the question marks in the subheadings and rephrased the sentences in the paragraph Conclusions and future therapeutic perspectives in the revised version of the manuscript (Lines 357-358 and 365-366).

Reviewer 2 Report

CKD management is one of the major health care issues in the world. Better understanding of the pathogenesis can lead to improved diagnostic and therapy options. In the proposed review, Yang and collegues summarize the most recent knowledge about kidney cell senescence and its possible contribution to kidney fibrosis and CKD in the aspect of protein-bound uremic toxins.

Here, I suggest some minor modifications and typo corrections. 

1) In the introduction, I would also present the pro- and contra results about renal EMT in vivo. The origin of myofibroblasts in kidney fibrosis could be activation of renal interstitial fibroblasts, pericytes, bone marrow-derived stem cells, endothelial cells (EndMT), apart of tubular epithelial cells (EMT). Although in vitro EMT can be easily induced in epithelial cells such as HK-2, the evidence for in vivo EMT mechanism in renal fibrosis is questionable. Please also cite the debating paper of Humphreys and colleagues (Humphreys BD, Lin SL, Kobayashi A, et al. Fate tracing reveals the pericyte and not epithelial origin of myofibroblasts in kidney fibrosis. Am. J. Pathol. 2010;176:85–97.) and some others.

2) In Line 64, TGF-beta signaling indeed can interact with other signaling pathways, please add here a list of a few (eg. MAPK / ERK, etc).

3) Line 103: "Reviewed in REF. 22" please delete REF.

4) Line 114: "(TIMP) is" should be "are".

5) In Figure 2, abbreviations of RS, OIS and SIS makes harder to follow for the reader, should be replaced by the full names (too many abbreviations in general do not improve the quality of reading)

6) Line 171 correct sentece such as "The balance between the pro-apoptotic and anti-apoptotic Bcl2-2 family members determines..."

7) In the text and figure legends, "toxicants" should be replace with "toxins"

8) Line 221: TGFb and CTGF are pro-fibrotic cytokines and not markers; also, IL-6 and IL-1b would be pro-inflammatory cytokine

9) Line 226: about CTGF, it is known to be the effector molecule of TGF-beta1 in the kidney as well - please add such reference here. Line 229: not only in lung epithelia, but also in the kidney (!) - here I would rather change this reference to kidney-related.

10) Line 233: IL-6 regulates MMPs and TGFb pathway - but how? Please add some examples and references.

11) Line 236: SASP factors regulate ECM production - please indicate how they regulate it!

12) Chapter 4 and 5 are in several parts redundant. I would delete chapter heading 4 and add its text into chapter 5, excluding the repetitions (eg line 257-259 is repeated in chapter 5 as well).

13) Line 293: if mTOR is involved in the pathogenesis, what is the effect of PBUTs on renal cell autophagy mechanisms? Dysfunction of autophagy is known to be a major contributor to CKD and fibrosis, possibly contributes to senescence too. Please add a few sentences about renal autophagy as well.

14) Line 314: grammar - change word order "the released inflammatory SASP factors..."

15) Line 316: "PBUTs accumulation-induced" is PBUT accumulation-induced

16) Line 355: "combatting kidney senescence" should be replaced with milder synonym like "ameliorating kidney senescence" 

Typos and minor grammar errors should be revised.

Author Response

We thank the reviewer for appreciating our work and for the valuable feedback, which helped us improving our manuscript. Below is our point-by-point response, modifications in our manuscript are given in red font:

1) We thank the reviewer for bringing up this question and the recommendation of the relevant references. Indeed, it should be clarified that the EMT mechanisms in renal fibrosis may differ between in vivo and in vitro experimental conditions. We included this explanation together with relevant references in the revised manuscript to improve the background introduction of the topic. (Lines 48 to 51).

2) to 8) We thank the reviewer for pointing out additional suggestions and corrections. We modified them in the revised version of the manuscript. (Lines 69, 106, 118, 177-178, 222 and 227).

9) We modified the description and added the references related to kidney. (Line 233-234).

10), 11) We thank the reviewer for the constructive suggestion. The example of how IL-6 regulates MMPs and TGFb pathway and how SASP regulate ECM production are now added in the revised manuscript. (Lines 244-248).

12) Chapters 4 and chapter 5 are now merged, and the repetitions were excluded in the revised manuscript.

13) We thank the reviewer for the constructive suggestions. We added the description about autophagy in kidney fibrosis and its possible relation to senescence. (Lines 306-311).

14) to 16) We modified the corrections in the revised version of the manuscript. (Lines 328, 330 and 368).